# SRSF2 is a key player in orchestrating the directional migration and differentiation of MyoD progenitors during skeletal muscle development

Rula Sha[1†], Ruochen Guo[1†], Huimin Duan[2], Qian Peng[1], Ningyang Yuan[1], Zhenzhen Wang[1], Zhigang Li[1], Zhiqin Xie[1], Xue You[2]*, Ying Feng[1,2]*

[1]CAS Key Laboratory of Nutrition, Metabolism and Food Safety, Shanghai Institute of Nutrition and Health, University of Chinese Academy of Sciences, Chinese Academy of Sciences, Shanghai, China; [2]Lin He's Academician Workstation of New Medicine and Clinical Translation in Jining Medical University, Jining Medical University, Jining, China

*For correspondence:
youxue19910@163.com (XY);
fengying@sinh.ac.cn (YF)

[†]These authors contributed equally to this work

Competing interest: The authors declare that no competing interests exist.

**Abstract** SRSF2 plays a dual role, functioning both as a transcriptional regulator and a key player in alternative splicing. The absence of Srsf2 in MyoD + progenitors resulted in perinatal mortality in mice, accompanied by severe skeletal muscle defects. SRSF2 deficiency disrupts the directional migration of MyoD progenitors, causing them to disperse into both muscle and non-muscle regions. Single-cell RNA-sequencing analysis revealed significant alterations in Srsf2-deficient myoblasts, including a reduction in extracellular matrix components, diminished expression of genes involved in ameboid-type cell migration and cytoskeleton organization, mitosis irregularities, and premature differentiation. Notably, one of the targets regulated by Srsf2 is the serine/threonine kinase Aurka. Knockdown of *Aurka* led to reduced cell proliferation, disrupted cytoskeleton, and impaired differentiation, reflecting the effects seen with *Srsf2* knockdown. Crucially, the introduction of exogenous Aurka in *Srsf2*-knockdown cells markedly alleviated the differentiation defects caused by *Srsf2* knockdown. Furthermore, our research unveiled the role of Srsf2 in controlling alternative splicing within genes associated with human skeletal muscle diseases, such as *BIN1*, *DMPK*, *FHL1*, and *LDB3*. Specifically, the precise knockdown of the *Bin1* exon17-containing variant, which is excluded following *Srsf2* depletion, profoundly disrupted C2C12 cell differentiation. In summary, our study offers valuable insights into the role of SRSF2 in governing MyoD progenitors to specific muscle regions, thereby controlling their differentiation through the regulation of targeted genes and alternative splicing during skeletal muscle development.

## eLife assessment

This **important** work provides interesting datasets of myofiber differentiation. The evidence supporting the involvement of SRF2 in selected biological processes is **convincing**, however, additional evidence to pin-point the major action of SRF2 during muscle differentiation is appreciated. The work will be of broad interest to developmental biologists in general and molecular biologists in the field of gene regulation.

## Introduction

Myogenic progenitors (MPCs) are a specialized type of precursor cell responsible for the formation of skeletal muscle cells during embryonic development. The process begins with the transformation of the dermomyotome into segmented structures known as somites, which serve as a reservoir for

MPCs expressing essential transcription factors Pax3 and Pax7 (*Relaix et al., 2006*; *Relaix et al., 2005*). These MPCs acquire their myogenic identity by activating specific myogenic regulatory factors, including Myf5, MyoD, Myog, and Myf6 (*Braun and Gautel, 2011*).

Among these factors, Myf5 + progenitors are key players in expanding the myogenic progenitor pool through extensive proliferation, as well as in the initial migration from somites to muscle-forming regions and the establishment of early muscle primordia (*Tajbakhsh et al., 1996*). The absence of Pax3 and Myf5 results in the inactivation of MyoD, ultimately leading to the absence of skeletal muscles in the body (*Tajbakhsh et al., 1997*). This highlights the hierarchical relationship between Myf5 and MyoD, with Myf5 acting as an upstream regulator of MyoD. Myf5 +progenitors serve as a source for MyoD + progenitors, these two populations collaborate in a coordinated manner to ensure the proper formation and growth of skeletal muscles during embryonic development (*Rodriguez-Outeiriño et al., 2021*). In the perinatal stages, MPCs play a crucial role in establishing the satellite cell (SC) pool, which serves as a reservoir of muscle stem cells (*Relaix et al., 2005*). MyoD was not expressed in quiescent SCs but showed high levels of expression in activated and proliferative myoblasts, which are the progeny of SCs (*Zammit et al., 2004*).

The extracellular matrix (ECM) and cytoskeleton play vital roles in maintaining the structural integrity of cells and tissues, while also exerting a profound influence on cell behavior (*Csapo et al., 2020*; *D'Urso and Kurniawan, 2020*). Specifically, the ECM serves as a scaffold for cell adhesion and migration, playing a critical role in regulating signaling pathways that control cell proliferation, differentiation, and survival (*Csapo et al., 2020*). On the other hand, the cytoskeleton is responsible for maintaining cell shape and polarity, facilitating intracellular transport, and aiding in cell division (*D'Urso and Kurniawan, 2020*). In the context of skeletal muscle development, interactions of MPCs with the ECM, as well as the dynamic rearrangements of the cytoskeleton, are critical for guiding the directional migration of MPCs (*Choi et al., 2020*; *Yin et al., 2013*). Actin filaments, a fundamental component of the cytoskeleton, undergo dynamic changes leading to the formation of protrusions like lamellipodia and filopodia, which assist in forward movement (*Lappalainen et al., 2022*). Meanwhile, microtubules, another critical part of the cytoskeleton, contribute to cell polarity and play a role in establishing and maintaining direction during migration. The reduction of both ECM and cytoskeleton components has a significant impact on myoblast migration, cell division, and differentiation (*Osses et al., 2009*; *Zhang et al., 2021*).

It's noteworthy that splicing and transcription regulator SRSF2 emerges as a critical regulator that inhibits apoptosis in My5 + progenitors and maintains their migratory capability during embryonic skeletal muscle development. The absence of Srsf2 in Myf5 + progenitors leads to a high rate of myoblasts apoptosis, as well as aberrant migration, ultimately resulting in the complete loss of mature myofibers (*Guo et al., 2022*). This emphasizes the essential role of Srsf2 in promoting the survival of Myf5 + progenitors and its significance in ensuring the successful progression of skeletal muscle development.

In this study, we observed that the absence of Srsf2 in MyoD + progenitors resulted in the perinatal mortality in mice, primarily due to respiratory failure. *Srsf2* deficiency markedly disrupted the directional migration capacity of MyoD + progenitors, causing them to disperse into both muscle and non-muscle regions. Single-cell RNA-sequencing (scRNA-seq) analysis revealed notable changes in Srsf2-deficient myoblasts, including reduced ECM components, diminished expression of genes involved in ameboid-type cell migration and cytoskeleton organization, mitosis irregularities, and impaired differentiation. Significantly, Srsf2 regulates the expression of the serine/threonine kinase Aurka. Knockdown of *Aurka* resulted in decreased cell proliferation and impaired differentiation, resembling *Srsf2* knockdown effects. Aurka overexpression in *Srsf2*-depleted cells significantly rescued the differentiation defects, underscoring the critical role of Srsf2-regulated *Aurka* expression in myoblast differentiation. Furthermore, our findings demonstrated the involvement of Srsf2 in alternative splicing of genes linked to human skeletal muscle diseases, including *BIN1*, *DMPK*, *FHL1*, and *LDB3*. Specifically, targeted knockdown of the *Bin1* exon17-containing variant, skipped upon *Srsf2* depletion, severely disrupted myoblast differentiation. In summary, our study provides valuable insights into SRSF2's role in directing MyoD progenitors to distinct muscle regions, thereby regulating their differentiation through targeted gene regulation and alternative splicing during skeletal muscle development.

# Results

## Inactivation of *Srsf2* in MyoD+ progenitors led to significant differentiation defects in skeletal muscle, resulting in perinatal death in mice

The absence of *Srsf2* in Myf5 + progenitors results in a significant increase in myoblasts apoptosis during embryonic development. This genetic condition leads to rib deformities and muscle loss in the limbs and body wall in mice, causing the demise of knockout pups immediately after birth (*Guo et al., 2022*). To eliminate the potential influence of rib deformities on mouse mortality and to concentrate on Srsf2's role in myogenesis, we employed a selective disruption of the *Srsf2* gene in MyoD + myogenic progenitors through crossbreeding *Srsf2*$^{fl/fl}$ mice (WT) with *Myod1*$^{Cre}$ mice (*Chen et al., 2005*), leading to the generation of *Srsf2*$^{fl/fl}$; *Myod1*$^{Cre}$mice (*Srsf2*-KO) (*Figure 1—figure supplement 1*).

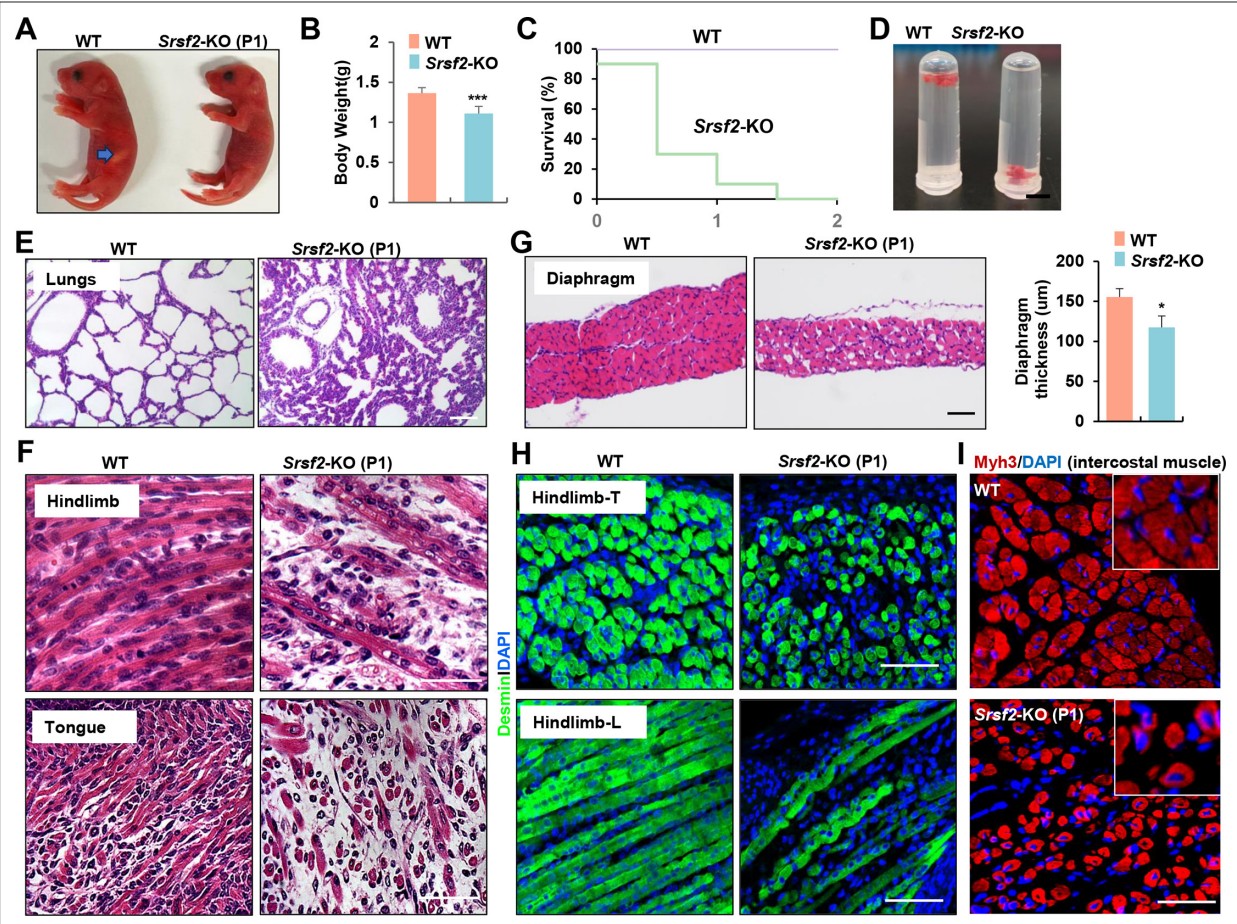

**Figure 1.** Inactivation of *Srsf2* in MyoD + progenitors resulted in skeletal muscle dysfunction and respiratory failure. (**A**) Macroscopic appearance of wild-type (WT) and *Srsf2*-KO pups immediately after birth (P1). Scale bars, 5 mm. (**B**) Comparison of body weight between WT and KO pups (n=10). (**C**) Survival curve of WT and KO pups (n=10). (**D**) Sinking test for lungs excised from P1 WT and KO mice and placed in a 5 ml Ep tube. (**E**) Representative HE staining of lung sections at P1 (n=3). Scale bars, 50 μm. (**F**) Representative HE staining on longitudinal sections of hindlimbs (top panels), and transverse tongue sections at P1 (bottom panels) (n=3). Scale bars, 10 μm. (**G**) Representative HE staining of transverse sections of diaphragms at P1 (n=3). Scale bars, 50 μm. Comparison of diaphragm thickness is displayed on the right. (**H**) Representative confocal images of desmin staining in transverse sections and longitudinal sections of hindlimbs from WT and KO at P1 (n=3). Scale bars, 10 μm. (**I**) Representative confocal images of Myh3 staining in intercostal muscle sections of WT and KO at P1 (n=3). Scale bars, 10 μm. Enlarged views are shown in the upper right corners.

The online version of this article includes the following source data and figure supplement(s) for figure 1:

**Figure supplement 1.** Schematic diagram of Srsf2-KO mice and genotyping.

**Figure supplement 1—source data 1.** Original files for the genomic PCR analysis in *Figure 1—figure supplement 1B* (floxed-allele and *Myod1*$^{Cre}$).

**Figure supplement 1—source data 2.** PDF containing annotation of original PCR gels in *Figure 1—figure supplement 1B* (floxed-allele and *Myod1*$^{Cre}$).

Newborn KO pups exhibited normal morphology but had empty stomachs without any milk and showed reduced body weight compared to their WT counterparts (*Figure 1A and B*). KO pups did not survive beyond the second day after birth (*Figure 1C*), suggesting potential respiratory issues. This observation prompted us to examine the condition of their lungs. The lungs of KO pups appeared congested compared to those of WT pups, and they sank in the water while WT lungs floated (*Figure 1D*). HE staining further revealed collapsed alveoli in the KO lungs at postnatal day 1 (P1) (*Figure 1E*).

Histological examination of various skeletal muscle regions revealed striking differentiation defects in P1 KO mice. Notably, the characteristic myofibers typically observed in the longitudinal sections of WT controls were conspicuously absent, instead replaced by shortened, irregular fibers in KO mice (*Figure 1F*, top panel). These muscle abnormalities were consistently observed in the tongue and diaphragm as well (*Figure 1F*, bottom panel). In transverse sections of the diaphragm from KO mice, there was a noticeable reduction in diaphragm thickness and the presence of cavities within the muscle fibers (*Figure 1G*). Desmin, a marker for myoblasts and newly formed fibers, exhibited significantly weaker staining in hindlimb sections of the KO group compared to the control group (*Figure 1H*). Myh3, a marker for terminal differentiation, displayed an irregular and loosened staining pattern along with the presence of central nuclei features in intercostal muscle sections of the KO group (*Figure 1I*). In summary, the deficiency of *Srsf2* had a profound and disruptive impact on the

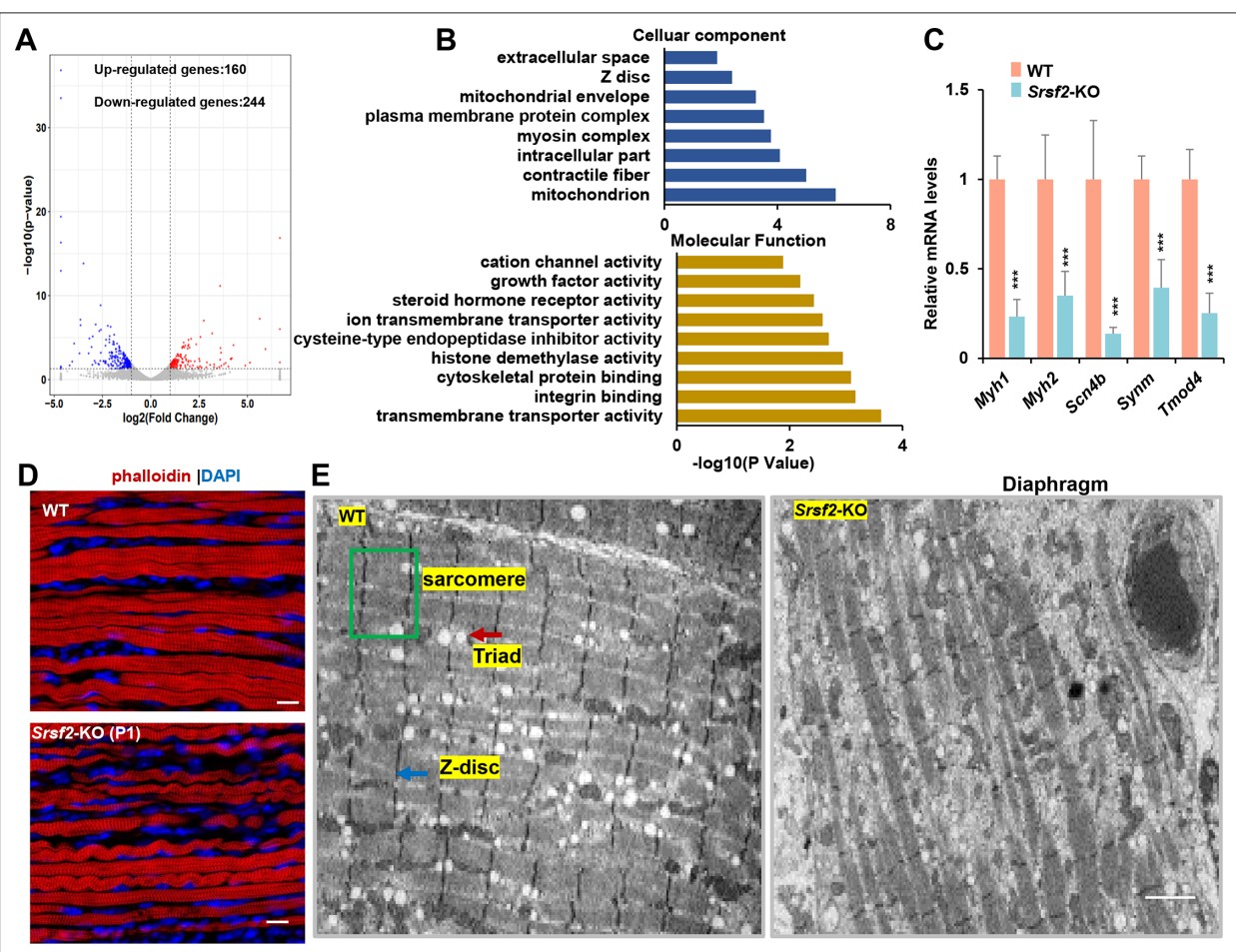

**Figure 2.** RNA-seq analysis and disrupted sarcomere structures in the KO skeletal muscle. (**A**) RNAs-seq was conducted on P1 diaphragm samples. Based on statistical significance (p-value <0.05) and fold change, over 160 upregulated genes and 244 downregulated genes were identified. (**B**) Gene Ontology (GO) enrichment analysis of differentially expressed genes (DEGs) was performed, focusing on their cellular component and molecular function categories. (**C**) qPCR analysis on a selection of genes based on RNA-seq data, using P1 diaphragm samples (n=6). (**D**) Representative confocal images of phalloidin staining in longitudinal sections of P1 diaphragm muscle of WT and *Srsf2*-KO mice. Scale bars, 10 μm. (**E**) Transmission electron microscopy images of the diaphragm in newborn WT and KO mice. Sarcomere is indicated by the green box, Z-disc by the blue arrow, and Triad by the red arrow. Scale bars, 20 μm.

formation of skeletal muscles, resulting in dysfunction that rendered KO mice unable to suckle for milk and breathe effectively.

## RNA-seq analysis and disrupted sarcomere structures in KO skeletal muscle

To explore the molecular mechanisms underlying *Srsf2* deficiency in skeletal muscle differentiation, we conducted RNA-seq analysis on diaphragm muscle samples collected from P1 mice. Differential gene analysis identified 160 up-regulated genes and 244 down-regulated genes (*Figure 2A*). Gene Ontology (GO) analysis showed that these differentially expressed genes (DEGs) were significantly enriched in key cellular components, including mitochondria, myosin complex, contractile fibers, Z-disc, among others. Moreover, these DEGs displayed enrichment crucial molecular functions such as transmembrane transporter activity, integrin binding, cytoskeletal protein binding, and among others (*Figure 2B*).

To validate these findings, we selected several genes based on their functions. For example, *Myh1* and *Myh2*, encode myosin-heavy polypeptides 1 and 2, respectively, while *Scn4b* encodes a protein associated with voltage-gated sodium channels. *Synm* plays a key role in the formation of sarcomeres, which are the contractile units of muscle cells, and Tmod4 is responsible for maintaining the structural integrity of skeletal muscle by regulating the stability of actin filaments. Our qPCR analysis demonstrates a significant reduction in the expression levels of these selected genes in the KO group (*Figure 2C*).

Actin is a major component of the thin filaments in the sarcomeres, and phalloidin is a fluorescent compound that selectively binds to and labels F-actin (filamentous actin). Phalloidin labeling in diaphragm sections revealed disarrangement of actin filaments in the KO group when compared to the WT group (*Figure 2D*). To gain deeper insights, we conducted transmission electron microscopy experiments to examine the sarcomere and Triad structures in the hindlimb sections. Sarcomeres are characterized by their highly organized structure, while Triads are specialized structures involved in the regulation of calcium ions required for muscle contraction. Our findings revealed well-organized sarcomere and Triad structures in the WT group. In contrast, these structures were substantially disrupted in the KO group (*Figure 2E*). This disruption could lead to impaired muscle functions, affecting their ability to contract and relax efficiently.

## *Srsf2* deficiency impairs the directional migration of MyoD+ progenitors and their subsequent differentiation during skeletal muscle development

We then developed a tracer mouse model through the crossbreeding of Rosa26-tdTomato mice with *Myod1*[Cre] and *Srsf2*[fl/fl] mice (*Figure 3A*). This approach allowed us to monitor and track the destiny of MyoD-derived cells, those that had previously exhibited MyoD expression or were currently expressing MyoD throughout the critical phases of skeletal muscle development.

In the analysis of embryo sections harvested from control mice at E14.5, the utilization of tdT staining, MHC staining, and the subsequent examination of the co-localization of tdT and MHC signals provided unequivocal evidence of the successful migration of MyoD-derived cells into the muscle region (*Figure 3B and C*; *Figure 3—figure supplement 1*). Subsequently, these cells underwent proliferation and differentiation, ultimately culminating in their significant contributions to the development of a highly organized muscle structure. In *tdT*/KO sections, MyoD-derived cells not only migrated into the intended muscle region but also dispersed widely into numerous non-muscular territories (*Figure 3B*, white arrows). This intriguing pattern strongly suggests migration defects of MyoD-derived cells resulting from the loss of Srsf2 at this stage. Moreover, even within the confines of the muscle region, *tdT*/KO embryos displayed a marked reduction in the degree of cellular differentiation when compared to the control (*Figure 3C*). This impaired differentiation ultimately led to a noticeable lack of precise co-localization between tdT and MHC signals (*Figure 3—figure supplement 1*).

Consistent with the migratory defects observed in E14.5 KO sections, in the hindlimb sections of E15.5 KO mice, as well as the hindlimb and intercostal muscle sections of KO mice at the perinatal stage (P1), tdT staining also revealed a substantial presence of MyoD-derived cells distributed beyond the muscle regions (*Figure 3D–F*, white arrows). These findings further confirmed that the deficiency



**Figure 3.** Srsf2-deficient MyoD-derived cells migrated into non-skeletal muscle regions and displayed significant differentiation defects. (**A**) A schematic diagram of *Srsf2*-KO/*tdT* mice (*Srsf2*^fl/fl^;*Myod1*^Cre^;*tdT*) was generated through the breeding of *Srsf2*^fl/fl^ mice with *Myod1*^Cre^ mice and *Rosa26-tdTomato* mice. *Srsf2*^fl/w^; *Myod1*^Cre^;*tdT* mice served as the control group. (**B**) Representative confocal images of tdT (red) staining on sagittal sections from E14.5 control and KO/tdT embryos. Scale bar is 500 μm. The red boxes indicated the enlarged images on the right, with a scale bar of 100 μm. (**C**) Representative confocal images of MHC (green) staining on sagittal sections from E14.5 control and KO/tdT embryos. Scale bars, 500 μm. White arrows indicated non-skeletal muscle regions. The enlarged images were shown on the right, with a scale bar of 100 μm. (**D**) Representative confocal images of tdT (red) staining in hindlimb sections from E15.5 mice. Scale bars, 100 μm. White arrows indicated non-skeletal muscle regions.

*Figure 3 continued on next page*

*Figure 3 continued*

(**E**) Representative confocal images of tdT (red) staining in hindlimb sections from P1 mice. Scale bars, 100 μm. White arrows indicated non-skeletal muscle regions. (**F**) Representative confocal images of tdT staining in transverse sections of intercostal muscle isolated from P1 mice. Scale bars, 200 μm. Enlarged images were shown on the left with a scale bar of 50 μm. (**G**) Representative confocal images of tdT (red) and MHC staining on longitudinal diaphragm sections from P1 mice. Scale bars, 20 μm. White arrows indicate undifferentiated cells or tdT-labeled cells without MHC staining. (**H**) Representative confocal images of tdT (red) and Laminin (green) immunostaining in transverse diaphragm sections from P1 mice. Scale bars, 20 μm. (**I**) Representative confocal images of tdT (red) and Laminin (green) immunostaining in hindlimb sections from P1 mice. Scale bars, 50 μm. White arrows highlight cavities within tdT/Lamin labeled cells.

The online version of this article includes the following figure supplement(s) for figure 3:

**Figure supplement 1.** Merged images of tdT and MHC on the sagittal sections of E14.5 embryos.

in SRSF2 profoundly impacts the capacity of MyoD-derived cells to undergo directed migration specifically into the skeletal muscle region.

Moreover, we conducted immunostaining on both longitudinal and cross-sections of the diaphragm in newborn mice (P1). In the control group, our examination of longitudinal sections revealed a remarkable co-localization between the tdT signals and the densely compacted muscle fibers stained with MHC (**Figure 3G**). Within the cross-sectional views, the tdT-stained regions exhibited a consistent and well-organized arrangement, surrounded by laminin (**Figure 3H**). In contrast, the KO mice exhibited noticeable gaps between muscle fibers in the longitudinal sections of the diaphragm, with only tdT signals present within these loosely arranged fibers (**Figure 3G**, white arrows). Furthermore, the tdT-stained areas appeared disorganized within the cross-sectional views, characterized by numerous cavities within them (**Figure 3H**, white arrows). The presence of similar cavities was also observed in the hindlimb sections of P1 KO mice (**Figure 3I**, white arrows). These collective observations underscore the critical role of SRSF2 in regulating both the migration and differentiation processes of MyoD-derived cells during muscle development.

## scRNA-seq analysis reveals distinct cell populations within the diaphragm

To further investigate the molecular mechanisms underlying the effect of Srsf2 in myogenesis, we extracted diaphragms from newborn WT and *Srsf2*-KO mice, prepared single-cell suspensions, and conducted scRNA-seq using the 10 X Genomics Chromium system (**Figure 4A**). After rigorous quality control (**Figure 4—figure supplement 1**), we obtained a final dataset consisting of 9506 WT cells and 11384 *Srsf2*-KO cells for downstream analysis. The resulting t-distributed stochastic neighbor embedding (t-SNE) analysis revealed 23 distinct cell clusters (**Figure 4—figure supplement 2**). Utilizing the top 10 marker genes in each cell cluster, correlation analysis between different cell clusters (**Figure 4—figure supplement 3**), and previously reported marker gene analysis, we successfully categorized the various cell types present in the diaphragm. These included fibroblasts, endothelial cells, skeletal muscle cells (SKM), mesothelial cells, immune cells, smooth muscle cells, tendon tenocytes, and glial cells (**Figure 4B–D**).

Evaluation of the cell composition proportions revealed no significant differences between WT and KO groups (**Figure 4E**). Individual t-SNE plots demonstrated a notable reduction in *Srsf2* expression specifically in SKM cells, while other cell types within the diaphragm of the *Srsf2*-KO group maintained similar levels of *Srsf2* compared to the control (**Figure 4F**).

## Precocious differentiation was observed in the SRSF2-deficient SKM population

We then conducted unsupervised cluster analysis specifically on the SKM population, which led us to identify six subclusters (sC1 to sC6) with distinct transcriptomes (**Figure 5A**; **Figure 5—figure supplement 1**). As depicted in **Figure 5B**, sC1 and sC2 exhibited similar levels of expression for the SC markers *Pax7* and *Myf5*. Notably, sC2 cells displayed elevated expression of *Myod1* and the proliferation marker *Mki67*, indicating their identity as cycling myoblasts (Cycling MB). In contrast, sC1 cells displayed low levels of *Myod1* expression and lacked *Mki67*, classifying them as non-cycling satellite cells (SCs). sC3 and sC4 cells were identified as committed MB and myocytes, respectively, based on

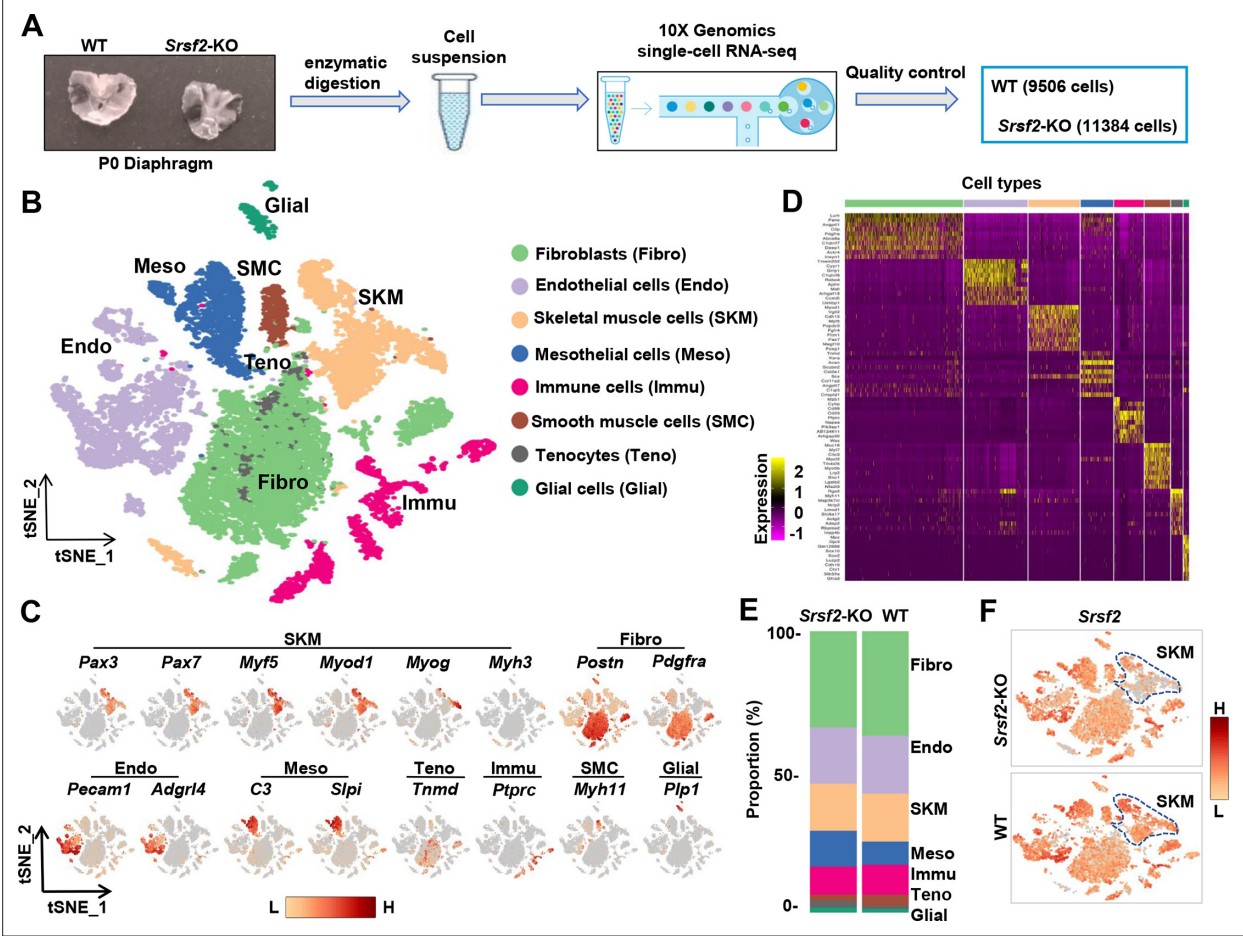

**Figure 4.** scRNA-seq analysis identifies distinct cell populations within P1 diaphragms. (**A**) Diagram of single-cell preparation from P1 diaphragms and processing using the 10 X Genomics platform for single-cell RNA-sequencing analysis. (**B**) t-SNE plot showing eight distinct cell populations identified in P1 wild-type (WT) and *Srsf2*-KO groups. Note that clusters expressing same marker genes were grouped into the same population. (**C**) t-SNE plot showing the expression of representative marker genes within distinct cell populations. (**D**) Heatmap showing the top 10 most differently expressed genes among eight distinct cell populations. (**E**) Proportion comparison of cell populations between WT and *Srsf2*-KO groups. (**F**) The levels of *Srsf2* are reduced in the *Srsf2*-KO skeletal muscle (SKM) cells compared to WT cells.

The online version of this article includes the following figure supplement(s) for figure 4:

**Figure supplement 1.** Analysis of scRNA-seq data.

**Figure supplement 2.** The t-distributed stochastic neighbor embedding (t-SNE plot) reveals 22 cell clusters that have been identified within the combined wild-type (WT) and *Srsf2*-KO diaphragm cells.

**Figure supplement 3.** Analysis of cell clusters based on sc-RNAseq data.

---

their expression of differentiation markers such as *Myog* and *Ckm*. For subsequent analysis, sC5 and sC6 cells were excluded due to their lack of distinct identity.

Subsequently, we employed Monocle2 (*Trapnell et al., 2014*) to investigate the dynamic cell transitions during myogenesis. Our analysis revealed that it organized the four subclusters into two major trajectories (*Figure 5C*). Cycling MBs (sC2) were positioned at the initiation of pseudotime trajectory, committed MB (sC3) and myocytes (sC4) were situated at the terminal phase of the differentiation branch, while non-cycling SCs (sC1) were positioned on the branch associated with quiescence. Our examination of dynamic gene expression profiles along the cycling to differentiation branch or the quiescence branch revealed distinct patterns (*Figure 5D*). Differentiation-related genes such as *Acta1 Ckm, Myh8, Tnnt1,* and *Tnnt3,* were highly expressed at the end of the differentiation branch (Module1). Proliferation markers including *Cdc20, Cdk1, Fbxo5, Mki67,* and *Prc1* were predominantly observed at the starting point of cycling (Module 2). QSC markers like *Gpx3, Myf5, Notch3, Hes1,* and *Pax7* were enriched at the end of the quiescence branch (Module 3). Genes such as *Cdkn1a, Myog,*

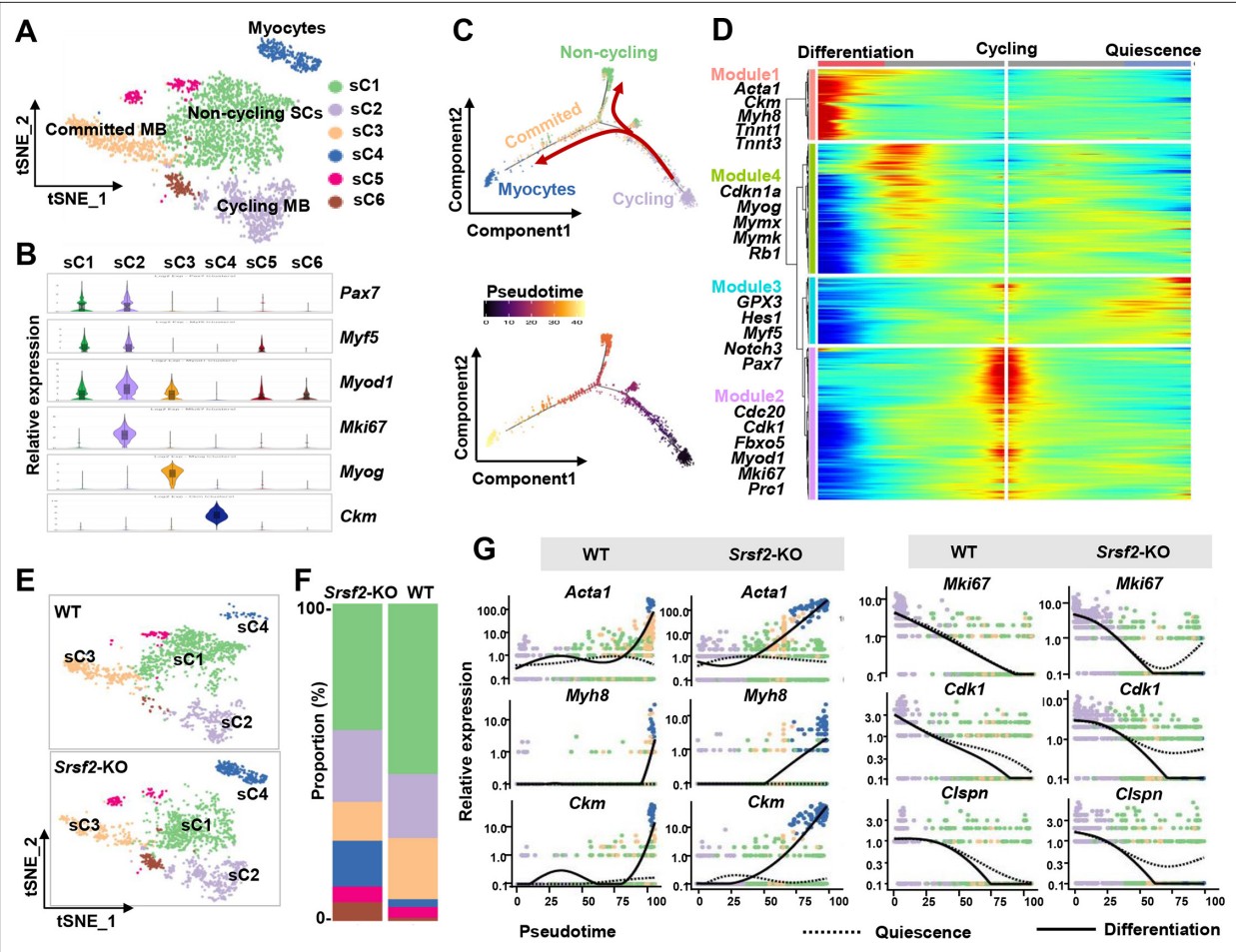

**Figure 5.** Loss of Srsf2 results in precocious differentiation of myoblasts. (**A**) The t-distributed stochastic neighbor embedding (t-SNE) plot reveals six distinct subclusters from wild-type (WT) and *Srsf2*-KO skeletal muscle cells (SKM) cells. (**B**) Violin plot showing expression levels of the marker genes in the subclusters. (**C**) Pseudotime-ordered analysis of SKM subclusters, shown as arrangement of subclusters along the two branches of the pseudotime (top), and their pseudotime order (bottom). (**D**) Heatmap showing expression dynamics of marker genes along the pseudotime branches. Representative marker genes in the four expression modules along the pseudotime shown on the left to the heatmap. (**E**) Separate t-SNE subcluster projections of WT (top) and *Srsf2*-KO (bottom) SKM cells. (**F**) Proportion comparison of WT and *Srsf2*-KO SKM subclusters. (**G**) Plots showing the relative levels of marker genes of WT (left) and *Srsf2*-KO (right) along the differentiation and quiescence branches of the pseudotime, respectively.

The online version of this article includes the following figure supplement(s) for figure 5:

**Figure supplement 1.** Heatmap showing the top 10 most differently expressed genes between six subclusters of skeletal muscle cells (SKM) cells.

*Mymx, Mymk,* and *Rb1* were highly expressed during the middle stage transitioning from cycling to the differentiation branch (Module 4). Thus, cycling MBs during perinatal myogenesis have two potential pathways: differentiation or quiescence.

Through the examination of separate t-SNE plots of SKM subclusters (*Figure 5E*) and the analysis of proportions (*Figure 5F*), several significant observations emerged. First, the proportion of non-cycling SCs (sC1) was reduced in the SRSF2-KO group, and their distribution notably deviated from that of sC1 in the WT group. Furthermore, the proportion of committed MB (sC3) decreased, while the proportion of myocytes (sC4) increased in the KO group when compared to the control. Consistently, the expression of differentiation-related genes such as *Acta1, Myh8,* and *Ckm* occurred earlier expression along the differentiation branch in the KO group as opposed to the WT group (*Figure 5G*, left panel). In contrast, proliferation-related genes like *Mki67, Cdk1,* and *Clspn* showed no expression at the end of the quiescence branch in the WT group, whereas their expression remained relatively high in the KO group (*Figure 5G*, right panel). These findings collectively indicate a precocious differentiation process in the absence of Srsf2.

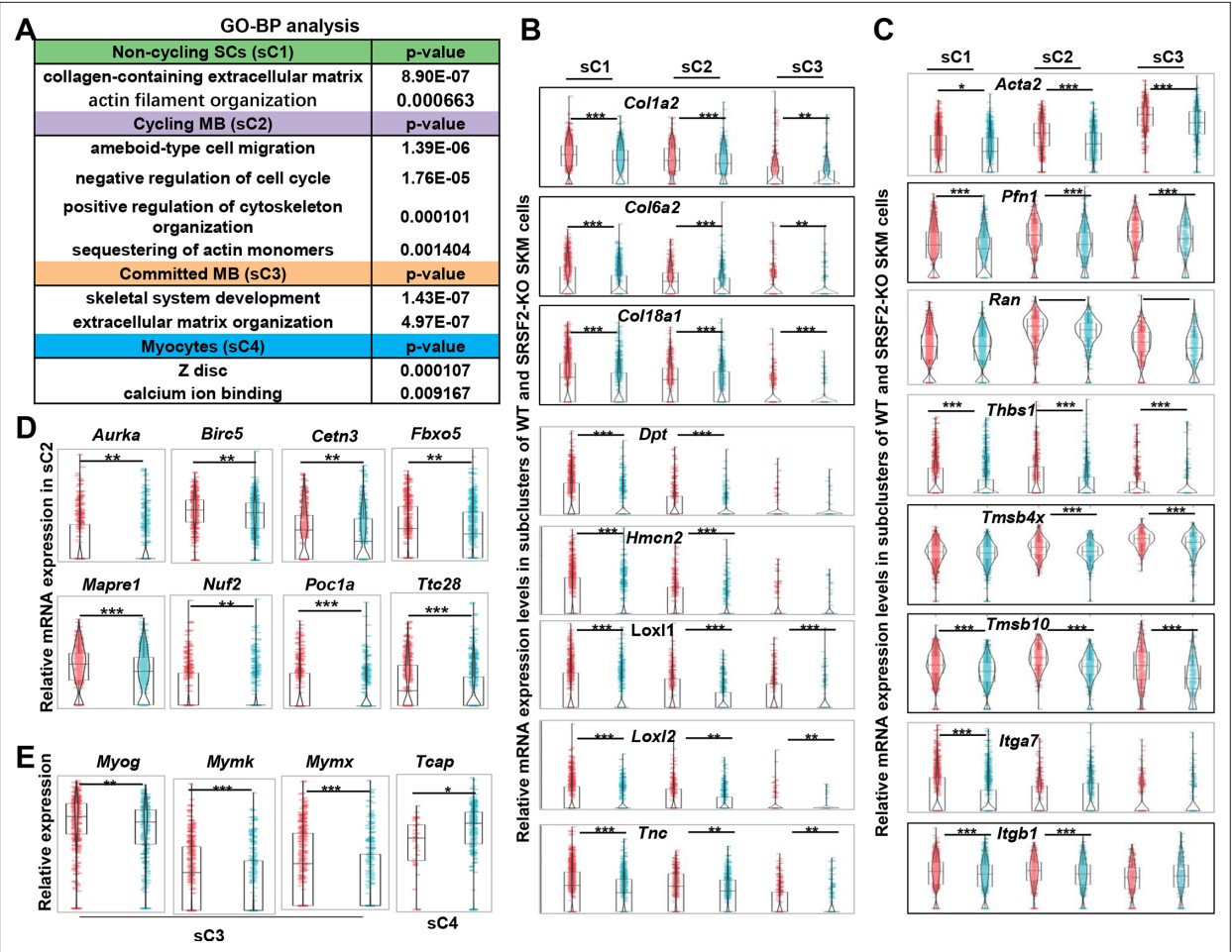

**Figure 6.** Decreased levels of extracellular matrix (ECM) and cytoskeleton, along with mitosis abnormalities and differentiation defects, were observed in the KO group. (**A**) Gene Ontology (GO) analysis of differentially expressed genes (DEGs) within four subclusters of skeletal muscle cells (SKM) cells. (**B**) Violin plots showing relative expression levels of ECM components between WT and KO subclusters. (**C**) Violin plots showing relative expression levels of cytoskeleton-related genes between WT and KO subclusters. (**D**) Violin plots showing relative expression levels of mitosis-related genes between WT and KO sC2 cells. (**E**) Violin plots showing relative expression levels of differentiation-related genes between WT and KO subclusters.

## Decreased ECM/cytoskeleton-related components, mitotic abnormalities, and defective differentiation were observed in the KO group

To further investigate the mechanisms underlying the observed precocious differentiation in KO myoblasts, we conducted an analysis of DEGs within subclusters, followed by a GO enrichment analysis. This analysis unveiled a substantial reduction in components associated with both the ECM and the cytoskeleton in the sC1, sC2, and sC3 cells of the KO group, as shown in *Figure 6A*. Notably, significant alterations were also observed in terms related to 'ameboidal-type cell migration' and 'negative regulation of cell cycle' within in the sC2 cells of the KO group. Additionally, there was a noteworthy reduction in genes involved in Z-disc formation within the sC4 cells of the KO group.

The ECM architecture is composed of complex supramolecular fibrillar networks that define tissue-specific cellular microenvironments. Genes encoding crucial collagen proteins for ECM structural integrity, such as *Col1a2*, *Col6a2*, and *Col18a1*, exhibited significant downregulation in sC1, sC2, and sC3 cells within the KO group (*Figure 6B*). Moreover, other ECM-related genes, including *Dpt* (dermatopontin), *Hmcn2* (hemicentin-2), *Loxl1* and *Loxl2* (lysyl oxidase-like 1 and 2), and *Tnc* (tenascin-C), all contributing to ECM composition and function, displayed significant downregulation in sC1, sC2, and sC3 of the KO group (*Figure 6B*).

In addition, genes associated with 'ameboidal-type cell migration' such as *Acta2* (actin alpha 2), *Pfn1* (profilin-1), *Ran* (Ras-related nuclear protein), and *Thbs1*(Thrombospondin-1) showed notable decreases in expression levels in sC1, sC2, and sC3 cells of the KO group. Genes related to cytoskeleton organization, such as *Tmsb4x* (thymosin beta-4) and *Tmsb10* (thymosin beta-10), were significantly downregulated in the KO group as well. Moreover, integrins, a family of cell adhesion receptors, mediating interactions between cells and the ECM, experienced significant downregulation in the KO group, particularly the expression of *Itga7* and *Itgb1*, integrin subunits (*Figure 6C*). This collective downregulation of ECM and cytoskeleton-related genes underscores the disruption in the molecular and structural interactions between myoblasts and their microenvironment in the KO group, potentially affecting cell migration, cell proliferation, and differentiation.

Indeed, individual genes encoding proteins play vital roles in regulating microtubule dynamics, spindle assembly, centriole and spindle pole formation, and mitotic progression control. Examples include *Aurka* (Aurora kinase A), *Birc5* (Survivin), *Cetn3* (Centrin-3), *Poc1a* (POC1 centriolar protein A), *Fbxo5* (F-box only protein 5), *Nuf2* (NDC80 kinetochore complex component Nuf2), *Mapre1* (Microtubule-associated protein RP/EB family member1), and *Ttc28* (Tetratricopeptide repeat domain 28) (*Figure 6D*). These genes displayed decreased expression in proliferating myoblasts (sC2) of the KO group, indicating a disruption in normal cell division and spindle assembly processes. *Mymk* (myomaker) and *Mymx* (myomerge) are essential for myoblast fusion, resulting in multinucleated myotube formation. Interestingly, their expression, along with *Myog*, was significantly downregulated in committed myoblasts (sC3) of the KO group (*Figure 6E*). On the other hand, *Tcap* (telethonin), involved in the assembly and maintenance of the Z-disc in skeletal muscle, showed increased expression in myocytes (sC4) of the KO group. This suggests the defective differentiation in the absence of SRSF1.

## SRSF2 regulates Aurka expression to control the differentiation of C2C12 myoblasts

Previous reports have demonstrated that SRSF2 functions as a potent transcription activator (*Cheng et al., 2016*; *Ji et al., 2013*). To investigate this further, we utilized a public database to search for SRSF2 ChIP-seq signals on genes that were downregulated in the Srsf2-deficient myoblasts, as revealed by scRNA-seq analysis. Among these genes, *Aurka*, pivotal in regulating cell cycle progression and microtubule dynamics, exhibited strong SRSF2 binding signals in its promoter regions (*Figure 7A*). ChIP-qPCR analysis provided additional evidence that SRSF2 directly binds to its promoter regions (*Figure 7B*). Moreover, the knockdown of Srsf2 had a significant impact on the reduction of Aurka protein levels in C2C12 myoblasts (*Figure 7C*, left panel). In contrast, when the *Aurka* gene was knocked down, it did not affect the protein levels of Srsf2 (*Figure 7C*, right panel), confirming that Srsf2 acts upstream of *Aurka*.

Intriguingly, both *Srsf2* and *Aurka* knockdowns resulted in decreased protein levels of cell-cycle- and proliferation-related genes such as Cdk1, pRb, p-pRb, p-Akt, while simultaneously increasing the expression of the cell-cycle inhibitor p27 and autophagy p62 (*Figure 7C*). Further analysis using RT-qPCR revealed that both knockdowns led to reduced levels of mitosis-related genes like *Fbxo5*, *Nuf2*, *Poc1a*, *Ranbp1,* and *Spdl1* (*Figure 7D and E*). Immunostaining analysis with phalloidin staining revealed both knockdowns induced significant alterations in the cytoskeleton and resulted in the formation of elongated pseudopods in the cells (*Figure 7F*).

Next, we aimed to investigate whether the knockdown of *Srsf2* and *Aurka* affects C2C12 cell differentiation. Following transfection of C2C12 cells with indicated siRNAs for 48 hr, cells were switched from growth medium to differentiation medium. Immunostaining with differentiation markers Myog and MHC revealed that both *Srsf2* and *Aurka* knockdowns significantly impeded the myogenic differentiation of C2C12 cells. While long multinucleated myotubes were observed in control cells, only short and unaligned myotubes were present following *SRSF2* or *Aurka* knockdowns (*Figure 7G and H*, lanes 1–6). Given that *Aurka* was an important target of Srsf2, we decided to test if Aurka overexpression could rescue differentiation defects seen in *Srsf2* depletion. To this end, we co-transfected C2C12 cells with siSRSF2 and an Aurka-overexpressing plasmid for 48 hr and then differentiated the cells for 3 days (*Figure 7H*, lanes 7–10). Overexpression of HA-tagged Aurka significantly rescued differentiation defects induced by downregulation of Srsf2, as evidenced by increased staining for Myog and MHC observed in Aurka-overexpressing cells (*Figure 7I*).



**Figure 7.** Srsf2 regulates *Aurka* expression to control mitotic processes and differentiation of C2C12 myoblasts. (**A**) UCSC genome browser screenshots illustrate SRSF2 ChIP-seq signals on the *Aurka* gene. A pair of primers is highlighted by red arrows, indicating their relative positions from the transcription start sites (TSS) within the *Aurka* promoter region on the top. (**B**) ChIP-qPCR analysis quantifies Srsf2-bound *Aurka* promoter regions in C2C12 myoblasts, with relative enrichment folds normalized to the IgG group. (**C**) Western blot analysis for indicated proteins in C2C12 cells transiently

*Figure 7 continued on next page*

*Figure 7 continued*

transfected with siRNAs against Srsf2 (siSrsf2-#1, siSrsf2-#2) for 48 hr (left) or against *Aurka* (siAurka-#1, siAurka-#2) for 48 hr (right). (**D**) qPCR analysis of spindle formation-related genes in C2C12 cells following *Srsf2* knockdown for 48 hr. (**E**) qPCR analysis of spindle formation-related genes in C2C12 cells with *Aurka* knockdown for 48 hr. (**F**) Immunostaining of phalloidin(red) and DAPI (blue) after *Srsf2* or *Aurka* knockdown. Scale bars, 20 μm. (**G**) Representative confocal images of MHC (green), Myog (red), and DAPI (blue) immunostaining in C2C12 cells. Scale bars, 50 μm. C2C12 cells were transfected with indicated siRNAs, then induced into differentiation for 3 days. Qualification of MHC + cells were shown on the right graphs. (**H**) Western blot analysis was performed using whole cell extracts isolated from C2C12 cells transfected with indicated siRNAs (lanes 1–6), or with the indicated siRNAs along with either vector plasmid or *Aurka* plasmid, respectively (lanes 7–10). (**I**) Representative confocal images of MHC (green), Myog (red), and DAPI (blue) immunostaining in C2C12 cells. Scale bars, 50 μm. C2C12 cells were co-transfected with the indicated siRNAs along with either vector plasmid or *Aurka* plasmid, respectively, then induced into differentiation for 3 days. Qualification of MHC + cells were shown on the right graphs.

The online version of this article includes the following source data for figure 7:

**Source data 1.** Original files for the western blotting analysis in *Figure 7C* (Srsf2, Aurka, CDK1, pRB, p-pRB, p27, AKT, p-AKT, p62, and Actin).

**Source data 2.** PDF containing annotation of original western blots in *Figure 7C* (Srsf2, Aurka, CDK1, pRB, p-pRB, p27, AKT,   p-AKT, p62, and Actin).

**Source data 3.** Original files for the western blotting analysis in *Figure 7H* (Srsf2, Aurka, and Actin).

**Source data 4.** PDF containing annotation of original western blots in *Figure 7H* (Srsf2, Aurka, and Actin).

## Srsf2 plays a crucial role in the regulation of alternative splicing in genes associated with human skeletal muscle diseases

SRSF2 serves a dual role, not only as a transcriptional regulator but also as a key player in alternative splicing. To explore its involvement in skeletal muscle development, we utilized the RNA-seq platform to investigate its role in the regulation of AS events. Our extensive analysis revealed a total of 23 AS events with significant changes in the KO muscle samples. Interestingly, several targets affected by Srsf2, such as *BIN1* (bridging interactor 1), *DMPK*(Dystrophin myotonica-protein kinase), *FHL1*(Four-and-a-half Lim domains protein 1), and *LDB3*(Lim domain binding 3), have been linked to conditions such as centronuclear myopathy, myotonic dystrophy, various X-linked muscle diseases, and myofibrillar myopathy (*Nicot et al., 2007*; *Pathak et al., 2021*; *Prokic et al., 2014*; *Thomas et al., 2018*; *Figure 8A*). Notably, the deficiency of *Srsf2* in the KO muscle samples led to exon 17 skipping in *Bin1*, exon5 skipping in *Fhl1*, exon 9 skipping in *Ldb3*, and the inclusion of the proximal 3'-alternative exon 14 in *Dmpk* (*Figure 8B*). These findings proved valuable insights into the potential role of Srsf2 in AS regulation and its connection to the development of human skeletal muscle diseases.

Nrap, Pdlim3, Pdlim7, and Trim54 are well-known for their contributions to muscle development and cytoskeletal structural integration (*D'Cruz et al., 2016*; *Jirka et al., 2019*; *Lodka et al., 2016*; *Yin et al., 2020*; *Figure 8A*). Notably, *Srsf2* deficiency resulted in specific alterations in their splicing patterns, including exon 12 skipping in *Nrap*, exon 5 skipping in *Pdlim3*, exon4 skipping of *Trim54*, and inclusion of exon 5 in Pdlim7 (*Figure 8C*). Genes associated with apoptosis and mitochondria, such as *Aamdc*, *Dap3* and *Dnm1l* (*Golden et al., 2021*; *Tong et al., 2023*; *Xiao et al., 2015*) were found to be under the splicing regulation of SRSF2. This led to exon4 skipping in *Aamdc* and exon 11 skipping in *Dap3*, while exon 17 was included in *Dnm1l* in the KO group (*Figure 8A and D*). Furthermore, *Ppp3ca* and *Ppp3cb* play vital roles in calcium handling in skeletal muscle cells (*Brinegar et al., 2017*). In the absence of *Srsf2*, exon 13 of *Ppp3ca,* and both exon 10 a and exon13 of *Ppp3cb* were skipped (*Figure 8A and D*). These findings strongly indicated that these dysregulated AS events play a contributing role in the effects of Srsf2 during skeletal muscle development.

Next, we aimed to analyze whether the knockdown of *Srsf2* affects these AS events in C2C12 cells. As illustrated in *Figure 8E*, *Srsf2* knockdown resulted in similar exon changes in genes such as *Bin1*, *Ldb3*, *Trim54*, *Dap3*, and *Dnm1l* in C2C12 cells as observed in KO skeletal muscles. Given that dysregulated splicing of *BIN1* is associated with muscle weakness in myotonic dystrophy (DM) (*Fugier et al., 2011*), we directed our focus toward *Bin1* variants for further analysis. As shown, *BIN1* contains 20 exons, with alternative exons regulated in different tissues (*Figure 8F*). Notably, exon 7 and exons 12-16 are excluded from human skeletal muscles, while exon 11 is excluded from normal brain tissues and DM skeletal muscles. However, we did not detect any alteration in exon 11 skipping in both *Srsf2*-knockdown cells and *Srsf2*-KO skeletal muscles (*Figure 8E and G*). Instead, our data demonstrated that Srsf2 is responsible for exon 17 inclusion in both skeletal muscles and C2C12 cells (*Figure 8B and E*). To investigate whether exon17 skipping of *Bin1* is involved in C2C12 differentiation, we designed siRNAs specifically targeting against *Bin1* exon17. Both siBin1-#1 and siBin1-#2 significantly decreased



**Figure 8.** Alternative splicing events affected by Srsf2 during skeletal muscle development. (**A**) A table displaying genes alternatively spliced by Srsf2 during skeletal muscle development. (**B**) RT-PCR analysis of AS events observed in *Bin1*, *Dmpk*, *Fhl1*, and *Ldb3* genes due to the absence of Srsf2, using total RNA isolated from three independent wild-type (WT) and KO samples. Alternative exons are highlighted in red with their respective exon number. Note that there were three 3' alternative sites within exon14 of *Dmpk*. Srsf2 deficiency resulted in selection of the proximal 3' site, while the

*Figure 8 continued on next page*

*Figure 8 continued*

presence of Srsf2 leads to the inclusion of the distal 3' site. (**C**) RT-PCR analysis of AS events observed in *Nrap*, *Pdlim3*, *Pdlim7,* and *Trim54* genes in three independent WT and KO skeletal muscle samples. (**D**) RT-PCR analysis of AS events observed in *Aamdc*, *Dap3*, *Dnm1l*, *Ppp3ca,* and *Ppp3cb* genes in three independent WT and KO skeletal muscle samples. (**E**) RT-PCR analysis of AS events observed in indicated targets following knockdown of *Srsf2* in C2C12 cells for 48 hr. (**F**) Diagram of *Bin1* splice variants present in both normal and diseased tissues. Note that green, red or gray exons represent alternative exons, while slight blue exons are constitutive. (**G**) RT-PCR analysis of *Bin1* exon 11 inclusion in three independent WT and KO skeletal muscle samples. (**H**) RT-PCR analysis was conducted to assess the levels of *Bin1* exon 17-containing and exon 17-lacking variants after transfection with siBin1-#1 or siBin1-#2 in C2C12 cells for 48 hr. The black arrows indicate primer positions. (**I**) Representative confocal images of MHC (green), Myog (red), and DAPI (blue) immunostaining in C2C12 cells. Scale bars, 50 μm. C2C12 cells were transfected with siBin1 for 48 hr, then induced into differentiation for 3 days. Quantification of MHC + cells were shown on the right graphs.

The online version of this article includes the following source data for figure 8:

**Source data 1.** Original files for RT-PCR analysis in *Figure 8B* (*Bin1*, *Dmpk*, *Phl1,* and *Ldb3*).

**Source data 2.** PDF containing annotation of original RT-PCR gels in *Figure 8B* (*Bin1*, *Dmpk*, *Phl1,* and *Ldb3*).

**Source data 3.** Original files for RT-PCR analysis in *Figure 8C* (*Nrap*, *Pdlim3*, *Pdlim7,* and *Trim54*).

**Source data 4.** PDF containing annotation of original RT-PCR gels in *Figure 8C* (*Nrap*, *Pdlim3*, *Pdlim7,* and *Trim54*).

**Source data 5.** Original files for RT-PCR analysis in *Figure 8D* (*Aamdc*, *Dap3*, *Dnm1l*, *Ppp3ca*, *Ppp3cb*-E10a, and *Ppp3cb*-E13).

**Source data 6.** PDF containing annotation of original RT-PCR gels in *Figure 8D* (*Aamdc*, *Dap3*, *Dnm1l*, *Ppp3ca*, *Ppp3cb*-E10a, and *Ppp3cb*-E13).

**Source data 7.** Original files for RT-PCR analysis in *Figure 8E* (*Bin1*-E17, *Ldb3*, *Trim54*, *Dap3*, *Dnm1l*, *Bin1*-E11, *Srsf2*, and *Rplp0*).

**Source data 8.** PDF containing annotation of original RT-PCR gels in *Figure 8E* (*Bin1*-E17, *Ldb3*, *Trim54*, *Dap3*, *Dnm1l*, *Bin1*-E11, *Srsf2*, and *Rplp0*).

**Source data 9.** Original files for RT-PCR analysis in *Figure 8G* (*Bin1*-E11).

**Source data 10.** PDF containing annotation of original RT-PCR gels in *Figure 8G* (*Bin1*-E11).

**Source data 11.** Original files for RT-PCR analysis in *Figure 8H* (*Bin1*-E17, and *Rplp0*).

**Source data 12.** PDF containing annotation of original RT-PCR gels in *Figure 8H* (*Bin1*-E17, and *Rplp0*).

the levels of exon17-containing variants in C2C12 cells (*Figure 8H*). The decreased staining for Myog and MHC strongly suggested impaired C2C12 differentiation following the specific knockdown of the *Bin1*-exon17 variant (*Figure 8I*). These findings highlighted the critical role of Srsf2-regulated exon17 inclusion of *Bin1* in C2C12 cell differentiation.

## Discussion

During embryonic skeletal muscle development, Myf5 + progenitors play a pivotal role in expanding the myogenic progenitor pool and migrating from somites to muscle-forming regions, establishing early muscle primordia [5b]. Notably, Srsf2 emerges as a critical regulator that inhibits apoptosis in My5 + progenitors and maintains their migratory capability during embryonic skeletal muscle development. The absence of Srsf2 in Myf5 + progenitors leads to a high rate of myoblasts apoptosis at embryonic day E12.5, as well as aberrant migration, ultimately resulting in the complete loss of mature myofibers.

MyoD + progenitors contribute to additional migration within muscle primordia, as they move to their final destinations within developing muscle tissues. They exhibit high efficiency in differentiation into myotubes and muscle fibers. Deletion of *Srsf2* in MyoD + progenitors markedly disrupts their ability to migrate in a specific direction and subsequently impairs their differentiation process.

scRNA-seq analysis revealed substantial diversity within the fetal skeletal muscle cell population in the diaphragm. This diversity included non-cycling SCs, activated myoblasts, cycling myoblasts, committed myoblasts, and myocytes. Our findings aligned with the observed diversity in muscle progenitors derived from human induced pluripotent stem cells (*Nalbandian et al., 2022*). Notably, the deficiency of *Srsf2* resulted in an increased number of myocytes while simultaneously reducing the population of committed myoblasts. In the context of skeletal muscle differentiation, Myh8, Ckm, and Acta1 are critical for the formation of muscle fibers (*Nowak et al., 2013*; *Schiaffino et al., 2015*). Both Myomaker and Myomerger proteins directly control the myogenic fusion process (*Chen et al., 2020*). However, the expression of *Myh8*, *Ckm,* and *Acta1* was significantly advanced at an earlier stage in the differentiation process, while there was a concurrent downregulation of *MyoK* and *Myox* in the absence of Srsf2. This suggests that enhanced differentiation was defective in the KO group.

scRNA-seq analysis further revealed significant effects of Srsf2 on the downregulation of ECM components, genes associated with ameboid-type cell migration, and cytoskeleton organization. The reduction of both ECM and cytoskeleton components has a significant impact on myoblast migration, cell division, and differentiation (*Osses et al., 2009*; *Zhang et al., 2021*). Notably, one of the targets regulated by Srsf2 is Aurka, a serine/threonine kinase that plays a crucial role in regulating cell division processes via the regulation of mitosis (*Du et al., 2021*). The chromosome passenger complex, which includes Aurka, inhibits the shrinkage of microtubules, thereby promoting cell growth (*Namgoong and Kim, 2018*). Our data revealed that the knockdown of *Aurka* led to reduced cell proliferation, disrupted cytoskeleton, and impaired differentiation, which mirrored the effects observed with the *Srsf2* knockdown. Importantly, overexpression of HA-Aurka in *Srsf2*-knockdown cells substantially rescued the differentiation defects observed following SRSF2 knockdown. These findings provided evidence that Srsf2-regulated *Aurka* expression is critical for differentiation of C2C12 cells.

In addition to its role in transcription regulation, SRSF2 also has a significant impact on the alternative splicing regulation of genes related to human skeletal muscle diseases. Dysregulated alternative splicing of *BIN1* has been linked to T-tubule alterations and muscle weakness in myotonic dystrophy (*Fugier et al., 2011*). BIN1 is involved in forming tubular of membrane invaginations and is critical for the development of muscle T-tubules, specialized membrane structures essential for excitation-contraction coupling. Mutations in the *BIN1* gene can lead to centronuclear myopathy, a condition that shares some histopathological features with myotonic dystrophy. MBNL1 binds the *BIN1* pre-mRNA and regulates its alternative splicing. Skipping of exon11 in BIN1 results in the expression of an inactive form of *BIN1* lacking phosphatidylinositol 5-phosphate-binding and membrane-tubulating activities. Notably, the deletion of *Srsf2* resulted in centronuclear features in skeletal muscle, and induced exon17 skipping in *Bin1*, which resides within the Myc-binding domain of Bin1 protein. Targeted knockdown of the *Bin1* exon17-containing variant significantly disrupted C2C12 cell differentiation. Further investigation is imperative to comprehend the involvement of SRSF2 and its regulation of exon17 inclusion in *BIN1* in human skeletal muscle diseases.

In summary, our research provides valuable insights into the role of Srsf2 in governing MyoD progenitors to specific muscle regions, thereby controlling myogenic differentiation through the regulation of targeted gens and alternative splicing during skeletal muscle development.

## Materials and methods
### Generation of conditional knockout mice
*Srsf2*^fl/fl mice were obtained from Dr. Fu at Westlake University. The *Srsf2*^fl/fl mice were bred with transgenic C57BL/6 J mice that express Cre recombinase under the control of the *Myod1* promoter. The resulting offspring (*Srsf2*^fl/w; *Myod1*^Cre) were then bred with *Srsf2*^fl/fl mice to generate *Srsf2* knockout mice *Srsf2*^fl/fl; *Myod1*^Cre (*Srsf2*-KO) The *Srsf2*^fl/fl /tdT mice used in this study were previously described (*Guo et al., 2022*). These mice were crossed with *Srsf2*^fl/w;*Myod1*^Cre mice to generate control mice (*Srsf2*^fl/w;*Myod1*^Cre; *tdT*) and *Srsf2*-KO/*tdT* mice (*Srsf2*^fl/fl;*Myod1*^Cre; *tdT* mice). All animal experiments were conducted in accordance with the guidelines of the Institutional Animal Care and Use Committee of the Shanghai Institute of Nutrition and Health, Chinese Academy of Sciences (approval no. SINH-2023-FY-1).

### HE staining and immunofluorescence staining
Mouse skeletal muscle tissues were fixed in 4% paraformaldehyde and embedded in paraffin or OCT. 7 µm-thick sections were stained with hematoxylin and eosin following standard protocols. For immunofluorescence on the paraffin-embedded tissues, 7µm-thick sections were deparaffinized in xylol for 30 minutes and rehydrated in a gradient of ethanol (100%, 95%, 75%, 50%) for 15 minutes, followed by antigen retrieval. The sections were then blocked with 5% normal goat serum (Santa Cruz Biotechnology) for 1 hr and incubated with primary antibodies overnight at 4°C. Subsequently, the secondary antibody was applied for 1 hr, and DAPI staining was performed for 10 minutes. For immunofluorescence on the OCT-embedded tissues, 5µm-thick sections were washed in PBS for 15 minutes to remove the OCT, and then the sections were blocked and stained similarly to the paraffin sections. Immunofluorescence images were acquired using a Zeiss LSM880NLO Confocal Microscope and the ZEN software. The detailed information on the antibodies used is listed in *Supplementary file 1*.

## Transmission electron microscopy

Diaphragms from P1 WT and *Srsf2*-KO mice were fixed overnight at 4 °C in a 2.5% glutaraldehyde solution, followed by four 15 minute PBS washes. Subsequently, they underwent dehydration in an ascending ethanol series (50%, 70%, 80%, and 90%) for 15 minutes each, with two rounds of 100% ethanol dehydration for 20 minutes each. The diaphragms were cleared twice with acetone for 15 minutes each and underwent a series of infiltrations. First, they were in an acetone: embedding medium (2:1 v/v) solution for 1 hr, followed by a solution of acetone: embedding medium (1:2 v/v) for 4 hr. Afterward, they were infiltrated twice with pure embedding medium for 20–24 hr each. After infiltration, the diaphragms were placed into a mold filled with pure embedding medium and polymerized at 65 °C for over 48 hr. The embedded block was trimmed into a trapezoid shape to ensure a sample surface area of less than 0.2mm x 0.2 mm. Ultrathin sections with 70 nm in thickness were cut and stained with uranyl acetate for 10 minutes, followed by lead citrate for another 10 minutes, with washing steps in between. Finally, the sections were examined under the transmission electron microscope.

## RNA-seq analyses

The total RNAs were extracted from the diaphragms of newborn mice and subjected to high-throughput transcriptome analysis using RNA sequencing (RNA-SEQ). Paired-end (PE) sequencing was performed using Next-Generation Sequencing (NGS) on the Illumina HiSeq sequencing platform. The raw data was processed using Cutadapt to trim 3' adapter sequences and reads with an average quality score below Q20 were removed, resulting in clean data. The clean data reads were then aligned to the mouse transcriptome (GRCm38/mm10) using HISAT2, an updated version of TopHat2. Gene expression levels were calculated based on the alignment results. Further analyses, including differential expression, enrichment, and alternative splicing analyses, were performed on the samples. DESeq was used for differential analysis of gene expression and GO enrichment analysis was performed using topGO. The alternative splicing analysis was performed with rMATs.

## Cell culture, differentiation, transfection, and immunostaining

C2C12 myoblasts were obtained from the National Collection of Authenticated Cell Cultures (Shanghai, China) and verified by STR analysis (Microread Genetics Co., Ltd, Beijing). Mycoplasma contamination was not detected in C2C12 cells. C2C12 myoblasts were cultured in DMEM supplemented with 10% fetal bovine serum. Transient transfection of C2C12 cells were performed using siRNA or plasmids. The *Aurka* overexpression plasmid was constructed using the longest NM isoform and inserted into the *pCDNA3.0-HA* plasmid vector. Myogenic differentiation of C2C12 cells was activated by replacing the growth medium with a differentiation medium (DMEM supplemented with 2% horse serum). The transfection was carried out using Lipofectamine RNAiMAX or Lipofectamine 200, both from Invitrogen, following the manufacturer's protocol. After 48 hr of transfection, or induction into differentiation for 3 days, the cells were fixed with 4% paraformaldehyde for 15 minutes. Then, permeabilization was done using 0.1% Triton X-100, and the cells were blocked with 5% normal goat serum for 1 hr. Subsequent staining was performed following the same protocol as tissue immunofluorescence staining. The siRNA sequences used for RNA interference can be found in *Supplementary file 1*.

## RNA extraction, RT-PCR, and qPCR

Total RNA was extracted from diaphragm tissues or C2C12 cells using Trizol Reagent (Invitrogen). The RNA was then reverse-transcribed into cDNA using the High-Capacity cDNA Reverse Transcription kit (Applied Biosystems). For differential expression analysis, qPCR was performed with the SYBR Premix Ex Taq kit (TaKaRa) with *Rplp0* serving as the internal control. For alternative splicing analysis, RT-PCR was conducted following the previously described method (*Cheng et al., 2016*; *Zhou et al., 2014*). The primer sequences used in this study are listed in *Supplementary file 1*.

## Preparation of single cell suspension and scRNA-seq

The diaphragms were isolated from P1 WT and KO mice. The diaphragms were cut to pieces, and then digested in a solution containing 0.2% (wt/vol) collagenase-type XI (Sigma-Aldrich) and 2.4 U mL⁻¹ dispase Ⅱ (Invitrogen) in DMEM at 37°C for 45 minutes. The digested tissue was filtered through

a 40 μm cell strainer and a 70 μm cell strainer. The cells were then centrifuged, resuspended in a 20% FBS/PBS buffer, and counted using an automated cell counter. The cells were loaded onto a droplet-based library prep platform Chromium (10 X Genomics) with a Chromium Single Cell Reagent Kit following the instructions provided in the protocol. Sequencing was performed on an Illumina Nova 6000 using 150-base pair paired-end reads. The sequencing and bioinformatics analysis were carried out by OE Biotech Co., Ltd. (Shanghai, China). All statistical analyses, unless otherwise specified, were performed using R.

## scRNA-seq data preprocessing

The Cell Ranger software pipeline (10 X Genomics, version 5.0.0) was utilized to assess data quality statistics on the raw data and map reads to the reference genome, resulting in a matrix of gene counts versus cells. Cells of low quality, such as double capture cells, multiple capture cells, or cells with more than 10% of counts from mitochondrial genes, were filtered out based on the nUMI, nGene, and percent. mito criteria using the R package Seurat (version 3.1.1). After applying these quality control measures, a total of 12,592 cells from the WT group and 14,875 cells from the *Srsf2*-KO group were available for further analysis. The library size was normalized using the NormalizeData function in Seurat, and dimensionality reduction was performed using the RunPCA function. Cell clusters were identified using the FindCluster in Seurat, and visualized using a two-dimensional t-distributed stochastic neighbor embedding (t-SNE) algorithm with the RunSNE function. Marker genes for each cluster were identified using the FindAllMarkers function, and cell types were manually determined based on cluster-specific marker genes and known marker genes. Differentially expressed genes (DEGs) were identified using the FindMarkers function (test.use= MAST) in Seurat (p-value <0.05 and | log2foldchange |>0.58). Additionally, GO enrichment and KEGG pathway enrichment analyses of DEGs were conducted using an R package based on the hypergeometric distribution.

## Pseudotime analysis

The developmental pseudotime was determined using the Monocle2 package (version 2.9.0) (*Trapnell et al., 2014*). The raw count was first converted from a Seurat object into a CellDataSet object using the importCDS function in Monocle. The differentialGeneTest function of the Monocle2 package was then used to select ordering genes (qval <0.01) that were likely to be informative in the ordering of cells along the pseudotime trajectory. Dimensional reduction clustering analysis was performed using the reduceDimension function, followed by trajectory inference with the orderCells function using default parameters. Gene expression was plotted using the plot_genes_in_pseudotime function to track changes over pseudo-time.

## Western blot analysis

For western blot analysis, proteins were collected from C2C12 cells using RIPA lysis buffer (Thermo Scientific) supplemented with a complete protease inhibitor cocktail (Roche). After quantification, equal amounts of proteins were subjected to immunoblotting. The primary antibodies used in this study were listed in *Supplementary file 1*.

## Chromatin immunoprecipitation assays (ChIP-qPCR)

ChIP-qPCR analysis was conducted to identify potential SRSF2 binding sites on the Aurka promoter regions. The biding site information was obtained from the Gene Expression Omnibus (GEO) database, specifically accession numbers GSM1106075 and GSM1106077. Two pairs of primers were designed based on this information for the ChIP-qPCR analysis. C2C12 cells were transiently transfected with *Srsf2-HA* and *vector-HA* plasmids for 48 hr. ChIP was performed using a Magna ChIP kit (Millipore; catalog no. MAGNA0017) following the manufacturer's instructions. Immunoprecipitation was carried out using anti-HA antibodies (Abcam), while normal mouse IgG served as a negative control. Quantification of the immunoprecipitated DNA and input DNA were done through qPCR analysis. Detailed information regarding the antibodies along with the primer sequences used in this study are listed in with *Supplementary file 1*.

## Statistical analysis

The data presented in the histograms were obtained from at least three independent experiments and are represented as the mean ± standard deviation (SD). In vivo data were obtained from at least three pups in each group. Statistical difference between groups was determined for using a two-tailed unpaired Student's t-test. p-value of *p<0.05, **p<0.01, and ***p<0.001 was considered statistically significant.

## Acknowledgements

This work was supported by grants from the National Key R&D Program of China (No. 2021 YFC2103001, for YF), the Shandong Province Natural Science Foundation (ZR2021QH151 for XY), the Research Fund for Lin He's Academician Workstation of New Medicine and Clinical Translation in Jining Medical University (JYHL2022MS26 for HD), the Shanghai Scientific Research Project (19JC1416000 for YF), and the National Natural Science Foundation (31870819 and 31570818 for YF). We thank Dr. Yongbing Ba and Lian Li (OE Biotech Co., Ltd, Shanghai, China) for their assistance with bioinformatics analysis.

# Additional information

## Funding

| Funder | Grant reference number | Author |
|---|---|---|
| National Key Research and Development Program of China | No. 2021YFC2103001 | Ying Feng |
| Natural Science Foundation of Shandong Province | ZR2021QH151 | Xue You |
| Jining Medical University | JYHL2022MS26 | Huimin Duan |
| Shanghai Scientific Research Project | 19JC1416000 | Ying Feng |
| National Natural Science Foundation | 31870819 | Ying Feng |
| National Natural Science Foundation | 31570818 | Ying Feng |

The funders had no role in study design, data collection and interpretation, or the decision to submit the work for publication.

## Author contributions

Rula Sha, Data curation, Formal analysis, Validation, Investigation, Visualization, Methodology, Writing - original draft; Ruochen Guo, Conceptualization, Formal analysis, Validation, Investigation, Visualization, Methodology; Huimin Duan, Resources, Funding acquisition, Visualization; Qian Peng, Ningyang Yuan, Zhenzhen Wang, Validation, Visualization, Methodology; Zhigang Li, Resources; Zhiqin Xie, Data curation, Project administration; Xue You, Resources, Funding acquisition, Visualization, Methodology; Ying Feng, Conceptualization, Formal analysis, Supervision, Funding acquisition, Writing - original draft, Writing - review and editing

## Author ORCIDs

Ying Feng ⓘ https://orcid.org/0000-0001-9501-156X

## Ethics

All animal experiments were conducted in accordance with the guidelines of the Institutional Animal Care and Use Committee of Shanghai Institute of Nutrition and Health, Chinese Academy of Sciences (approval no. SINH 2023 FY 1).

Reviewer #1 (Public Review): https://doi.org/10.7554/eLife.98175.2.sa1
Reviewer #2 (Public Review): https://doi.org/10.7554/eLife.98175.2.sa2
Author response https://doi.org/10.7554/eLife.98175.2.sa3

## Additional files

### Supplementary files
• Supplementary file 1. Antibody information and primer sequences are listed. (a) Antibodies used in IF, WB, and ChIP analysis. (b) Primer sequences used for RT-PCR. (c) Primer sequence used for qPCR. (d) SiRNA sequence used for RNA interference. (e) Primer sequence used for ChIP-qPCR.

• MDAR checklist

### Data availability
The raw RNA-seq datasets have been submitted to the BioProject database under accession number PRJNA974218. The scRNA-seq datasets have been submitted to the GEO database under accession number GSE233227. Source data for genomic PCR analysis, RT-PCR analysis and original western blots are available in source data files.

The following datasets were generated:

| Author(s) | Year | Dataset title | Dataset URL | Database and Identifier |
|---|---|---|---|---|
| Sha R, Feng Y | 2023 | RNA-sequencing of diaphragms from WT, SRSF2 knock-out mice | https://www.ncbi.nlm.nih.gov/bioproject/PRJNA974218 | NCBI BioProject, PRJNA974218 |
| Sha R, Wang Z, Feng Y | 2024 | Single-cell sequencing of diaphragms from both control and knockout mice at postnatal day 1 | https://www.ncbi.nlm.nih.gov/geo/query/acc.cgi?acc=GSE233227 | NCBI Gene Expression Omnibus, GSE233227 |

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
