## [Editor Report · eLife assessment]

This **important** work provides interesting datasets of myofiber differentiation. The evidence supporting the involvement of SRF2 in selected biological processes is **convincing**, however, additional evidence to pin-point the major action of SRF2 during muscle differentiation is appreciated. The work will be of broad interest to developmental biologists in general and molecular biologists in the field of gene regulation.

---

## [Referee Report · Reviewer #1 (Public Review)]

Summary

The work by She et al. investigates the role of SRFS2 in the MyoD+ progenitor cells during development. Deletion of SRFS2 in MyoD+ progenitor cells resulted in a defect in the directional migration of these cells and resulted in the presence of myoD+ progenitor in both nonmuscle and muscle tissues. The authors showed a defect in gene program regulation ECM, cell migration, cytoskeletal organization, and skeletal muscle differentiation by scRNA-seq. The authors further showed that many of these processes are regulated by a downstream target of SRFS2, the serine-threonine kinase Aurka. Finally, the authors showed that SRFS2 acts as a splicing factor and could contribute to differentiation by controlling the splicing of muscle-specific transcripts. This study addresses an important question in skeletal muscle development by focusing on the pathways and factors that regulate the migration of myoD+ progenitors and the impact of this process in skeletal muscle differentiation. This work is interesting but requires experimental evidence to support the findings.

Strengths

The regulators of myod+progenitor migration during skeletal muscle development is not completely understood. This work demonstrates that SRFS2 and aura kinase are key players in the process. Combining knockout and reporter lines in mice, the authors perform a detailed analysis of skeletal muscle cells to demonstrate the specific defects in SRFS2 in skeletal muscle development.

Weaknesses

This work explores an interesting question on regulating myoD+ progenitors and the defects of this process in skeletal muscle differentiation by SRFS2 but spreads out in many directions rather than focusing on the key defects. A number of approaches are used, but they lack the robust mechanistic analysis of the defects that result in muscle differentiation. Specifically, the role of SRFS2 on splicing appears to be a misfit here and does not explain the primary defects in the migration of myoD+ progenitors. There are concerns about the scRNA-seq and many transcripts in muscle biology that are not expressed in muscle cells. Focusing on main defects and additional experimental evidence to clear the fusion vs. precocious differentiation vs. reduced differentiation will strengthen this work.

(1) The analysis of RNA-seq data (Figure 2) is limited, and it is unclear how it relates to the work presented in this MS. The Go enrichment analysis is combined for both up and down-regulated DEG, thus making it difficult to understand the impact differently in both directions. Stac2 is a predominant neuronal isoform (while Stac3 is the muscle), and the Symm gene is not found in the HGNC or other databases. Could the authors provide the approved name for this gene? The premise of this work is based on defects in ECM processes resulting in the mis-targeting of the muscle progenitors to the nonmuscle regions. Which ECM proteins are differentially expressed?

(2) Could authors quantify the muscle progenitors dispersed in nonmuscle regions before their differentiation? Which nonmuscle tissues MyoD+ progenitors are seen? Most of the tDT staining in the enlarged sections appears to be punctate without any nuclear staining seen in these cells (Figure 3 B, D E-F). Could authors provide high-resolution images? Also, in the diaphragm cross-sections in mutants, tdT labeling appears to be missing in some areas within the myofibers defined as cavities by the authors (marked by white arrows, Figure 3H). Could this polarized localization of tDT be contributing to specific defects?

(3) Is there a difference in the levels of tDT in the myoD" muscle progenitors that are mis-targeted vs the others that are present in the muscle tissues?

(4) scRNA is unsuitable for myotubes and myofibers due to their size exclusion from microfluidics. Could authors explain the basis for scRNA-seq vs SnRNA-seq in this work? How are SKM defined in scRNA-data in Figure 4? As the myofibers are small in KO, could the increased level of late differentiation markers be due to the enrichment of these small myotubes/myofibers in scRNA? A different approach, such as ISH/IF with the myogenic markers at E9.5-10.5, may be able to resolve if these markers are prematurely induced.

(5) TNC is a marker for tenocytes and is absent in skeletal muscle cells. The authors mentioned a downregulation of TNC in the KO SKM derived clusters. This suggests a contamination of the tenocytes in the control cells. In spite of the downregulation of multiple ECM genes showed by scRNA-seq data, the ECM staining by laminin in KO in Figure 3 appears to be similar to controls.

(6) The expression of many fusion genes, such as myomaker and myomerger, is reduced in KO, suggesting a primary fusion defect vs a primary differentiation defect. Many mature myofiber proteins exhibit an increased expression in disease states, suggesting them as a compensatory mechanism. Authors need to provide additional experimental evidence supporting precocious differentiation as the primary defect.

(7) The fusion defects in KO are also evident in siRNA knockdown for SRSF2 and Aurka in C2C12, which mostly exhibits mononucleated myocytes in knockdowns. Also, a fusion index needs to be provided.

(8) The last section of the role of SRSF2 on splicing appears to be a misfit in this study. Authors describe the Bin1 isoforms in centronuclear myopathy, but exon17 is not involved in myopathy. Is exon17 exclusion seen in other diseases/ splicing studies?

---

## [Referee Report · Reviewer #2 (Public Review)]

Summary:

This study was aimed to study the role of SRSF2 in governing MyoD progenitors to specific muscle regions. The Results confirmed the role of SRSF2 in controlling myogenic differentiation through the regulation of targeted genes and alternative splicing during skeletal muscle development.

Strengths:

The study used different methods and techniques to achieve aims and support the conclusions such as RNA sequencing analysis, Gene Ontology analysis, immunostaining analysis.

This study provides novel findings that SRSF2 controls the myogenic differentiation of MyoD+ progenitors, using transgenic mouse model and in vitro studies.

Weaknesses:

Although unbiased sequencing methods were used, their findings about SRSF2 served as a transcriptional regulator and functioned in alternative splicing events are not novel.

The introductions and discussion is not clearly written. The authors did not raise clear scientific questions in the introduction part. The last paragraph is only copy-paste of the abstract. The discussion part is mainly the repeat of their results without clear discussion.

---

## [Author Response]

**Reviewer #1 (Public Review):**
[…] Weaknesses:This work explores an interesting question on regulating myoD+ progenitors and the defects of this process in skeletal muscle differentiation by SRFS2 but spreads out in many directions rather than focusing on the key defects. A number of approaches are used, but they lack the robust mechanistic analysis of the defects that result in muscle differentiation. Specifically, the role of SRFS2 on splicing appears to be a misfit here and does not explain the primary defects in the migration of myoD+ progenitors. There are concerns about the scRNA-seq and many transcripts in muscle biology that are not expressed in muscle cells. Focusing on main defects and additional experimental evidence to clear the fusion vs. precocious differentiation vs. reduced differentiation will strengthen this work.(1) The analysis of RNA-seq data (Figure 2) is limited, and it is unclear how it relates to the work presented in this MS. The Go enrichment analysis is combined for both up and down-regulated DEG, thus making it difficult to understand the impact differently in both directions. Stac2 is a predominant neuronal isoform (while Stac3 is the muscle), and the Symm gene is not found in the HGNC or other databases. Could the authors provide the approved name for this gene? The premise of this work is based on defects in ECM processes resulting in the mis-targeting of the muscle progenitors to the nonmuscle regions. Which ECM proteins are differentially expressed?

The GO enrichment analysis (Figure 2B) indicates that genes involved in skeletal muscle construction and function were significantly dysregulated, with both up-regulated and down-regulated genes observed, consistent with the phenotype analysis presented in Figure 1.

We agree with the reviewer’s comments that Stac3 is the predominant muscle isoform with high expression in skeletal muscle tissues, while stac2 is expressed at low levels in these tissues. Therefore, we decided to delete the Stac2 data from the Figure 2C and will modify the text accordingly. We apologize for our errors.

In response to the reviewer's comment regarding the Symm gene not being found in the HGNC or other databases, we carefully re-examined the genes presented in Figure 2C. We discovered that one of the genes is actually Synm, which encodes synemin, an intermediate filament protein. We will correct this in the manuscript.

scRNA-seq analysis revealed defects in ECM processes in SRSF2-deficient myoblasts, which we believe likely resulted in the mis-targeting of muscle progenitors to non-muscle regions. However, comparing RNA-seq results from whole muscle tissues with scRNA-seq results is challenging.

(2) Could authors quantify the muscle progenitors dispersed in nonmuscle regions before their differentiation? Which nonmuscle tissues MyoD+ progenitors are seen? Most of the tDT staining in the enlarged sections appears to be punctate without any nuclear staining seen in these cells (Figure 3 B, D E-F). Could authors provide high-resolution images? Also, in the diaphragm cross-sections in mutants, tdT labeling appears to be missing in some areas within the myofibers defined as cavities by the authors (marked by white arrows, Figure 3H). Could this polarized localization of tDT be contributing to specific defects?

tdT staining revealed a substantial presence of MyoD-derived cells distributed beyond the muscle regions, as shown in Figure 3B. Quantify the number of MyoD+ progenitors dispersed in non-muscle regions is not meaningful.

tdT+ cells also include those that previously expressed MyoD but have since differentiated into myotubes and myofibers, which is why many tdT+ staining is not nuclear.

MyoD+ cells deficient in SRSF2 either undergo apoptosis or premature differentiation. Consequently, tdT staining in SRSF2-KO muscles showed many irregularities in the muscle fibers.

(3) Is there a difference in the levels of tDT in the myoD" muscle progenitors that are mis-targeted vs the others that are present in the muscle tissues?

tdT+ cells include those that previously expressed MyoD but have since differentiated into myotubes and myofibers, which are no longer MyoD+ cells. Additionally, tdT+ also include those currently expressing MyoD, which are MyoD+ cells.

The fiber differences between WT and SRSF2-KO mice are easily discernible through tdT staining (Figure 2D and 3D), however, comparing the levels of tdT staining between the two groups is not meaningful.

(4) scRNA is unsuitable for myotubes and myofibers due to their size exclusion from microfluidics. Could authors explain the basis for scRNA-seq vs SnRNA-seq in this work? How are SKM defined in scRNA-data in Figure 4? As the myofibers are small in KO, could the increased level of late differentiation markers be due to the enrichment of these small myotubes/myofibers in scRNA? A different approach, such as ISH/IF with the myogenic markers at E9.5-10.5, may be able to resolve if these markers are prematurely induced.

SRSF2 is highly expressed in proliferative myoblasts, but its levels declined once differentiation begins. In our study, we used Myod1-Cre to delete the SRSF2 gene and performed the scRNA-seq analysis to examine the effects of SRSF2 deletion on the proliferation and differentiation of MyoD cells. Our analysis revealed that SRSF2 deletion caused proliferation defects and premature differentiation of MyoD cells (Figure 5G), leading to myofiber abnormalities.

We determined that snRNA-seq analysis is not suitable for our study.

Additionally, skeletal muscle cells (SKM) were defined based on the expression of skeletal muscle markers, as shown in Figure 4C.

(5) TNC is a marker for tenocytes and is absent in skeletal muscle cells. The authors mentioned a downregulation of TNC in the KO SKM derived clusters. This suggests a contamination of the tenocytes in the control cells. In spite of the downregulation of multiple ECM genes showed by scRNA-seq data, the ECM staining by laminin in KO in Figure 3 appears to be similar to controls.

Tenascin-C (Tnc) is also part of the extracellular matrix (ECM) family. scRNA-seq analysis revealed that multiple ECM genes were downregulated in SRSF2-KO myoblasts, however, this did not indicate that laminin was downregulated in the SRSF2-KO muscles.

(6) The expression of many fusion genes, such as myomaker and myomerger, is reduced in KO, suggesting a primary fusion defect vs a primary differentiation defect. Many mature myofiber proteins exhibit an increased expression in disease states, suggesting them as a compensatory mechanism. Authors need to provide additional experimental evidence supporting precocious differentiation as the primary defect.

Our analysis revealed that the deletion of SRSF2 caused premature differentiation of MyoD cells (Figure 5G), leading to abnormalities of myofiber formation. SRSF2 is highly expressed in proliferative myoblasts, but its expression declines quickly in myotubes. Therefore, it is unlikely that the low expression of SRSF2 in myotubes caused the primary fusion defect.

(7) The fusion defects in KO are also evident in siRNA knockdown for SRSF2 and Aurka in C2C12, which mostly exhibits mononucleated myocytes in knockdowns. Also, a fusion index needs to be provided.

SRSF2 knockdown and Aurka knockdown caused differentiation defects, including fusion defects. We quantified the percentages of both MyoG+ and MHC+ cells in the differentiation assay.

(8) The last section of the role of SRSF2 on splicing appears to be a misfit in this study. Authors describe the Bin1 isoforms in centronuclear myopathy, but exon17 is not involved in myopathy. Is exon17 exclusion seen in other diseases/ splicing studies?

Our study is the first to report that exon 17 inclusion of Bin1 is regulated by SRSF2. Specifically, the knockdown of Bin1 exon 17 caused severe differentiation defects in C2C12 myoblasts. The involvement of Bin1 exon 17 in myopathy requires further validation using clinical samples.

**Reviewer #2 (Public Review):**
[…] Weaknesses:Although unbiased sequencing methods were used, their findings about SRSF2 served as a transcriptional regulator and functioned in alternative splicing events are not novel. The introductions and discussion is not clearly written. The authors did not raise clear scientific questions in the introduction part. The last paragraph is only copy-paste of the abstract. The discussion part is mainly the repeat of their results without clear discussion.

While the role of SRSF2 as a transcriptional regulator involved in alternative splicing events is not novel, the specific SRSF2-regulated alternative splicing events and targeted genes in skeletal muscle have not been reported in other publications. We believe our interpretation of the data and comparison with related published studies are well presented in the Discussion section.